# Impact of the COVID-19 Outbreak—Delayed Referral of Colorectal and Lung Cancer in Primary Care: A National Retrospective Cohort Study

**DOI:** 10.3390/cancers15051462

**Published:** 2023-02-25

**Authors:** Charles W. Helsper, Carla H. Van Gils, Nicole F. Van Erp, Marinde F. R. Siepman van den Berg, Omar Rogouti, Kristel M. Van Asselt, Otto R. Maarsingh, Jean Muris, Daan Brandenbarg, Sabine Siesling, Niek J. De Wit, Matthew P. Grant

**Affiliations:** 1Julius Centre for Health Sciences and Primary Care, University Medical Centre Utrecht, Utrecht University, Heidelberglaan 100, 3584 CS Utrecht, The Netherlands; 2Department of General Practice, Amsterdam UMC, Amsterdam Public Health Research Institiute, Meibergdreef 9, 1105 AZ Amsterdam, The Netherlands; 3Department of General Practice, Care and Public Health Research Institute, Maastricht University, P. Debeyeplein 1, 6229 HA Maastricht, The Netherlands,; 4Department of General Practice and Elderly Care Medicine, University Medical Centre Groningen, University of Groningen, 9700 AD Groningen, The Netherlands; 5IKNL—Department of Research and Development, Netherlands Comprehensive Cancer Organisation (IKNL), Godebaldkwartier 419, 3511 DT Utrecht, The Netherlands; 6Department of Health Technology and Services Research, Utrecht & Technical Medical Centre, Twente University, Technohal, Hallenweg 5, 7522 NH Enschede, The Netherlands

**Keywords:** COVID, cancer diagnosis, primary care, delay

## Abstract

**Simple Summary:**

The COVID-19 pandemic impacted health care. We studied the impact of the COVID-19 outbreak on the referral and diagnosis of cancer patients, thereby aiming to help prevent the delay of cancer diagnosis in future pandemics. We found that during the COVID-19 outbreak, the time between presentation and referral in primary care for patients with colorectal and lung cancer was substantially increased. This was also seen for patients who presented with alarm symptoms. This shows that, in future pandemics, targeted GP support is needed to maintain effective cancer diagnosis.

**Abstract:**

The Coronavirus disease 2019 (COVID-19) outbreak impacted health care. We investigated its impact on the time to referral and diagnosis for symptomatic cancer patients in The Netherlands. We performed a national retrospective cohort study utilizing primary care records linked to The Netherlands Cancer Registry. For patients with symptomatic colorectal, lung, breast, or melanoma cancer, we manually explored free and coded texts to determine the durations of the primary care (IPC) and secondary care (ISC) diagnostic intervals during the first COVID-19 wave and pre-COVID-19. We found that the median IPC duration increased for colorectal cancer from 5 days (Interquartile Range (IQR) 1–29 days) pre-COVID-19 to 44 days (IQR 6–230, *p* < 0.01) during the first COVID-19 wave, and for lung cancer, the duration increased from 15 days (IQR) 3–47) to 41 days (IQR 7–102, *p* < 0.01). For breast cancer and melanoma, the change in IPC duration was negligible. The median ISC duration only increased for breast cancer, from 3 (IQR 2–7) to 6 days (IQR 3–9, *p* < 0.01). For colorectal cancer, lung cancer, and melanoma, the median ISC durations were 17.5 (IQR (9–52), 18 (IQR 7–40), and 9 (IQR 3–44) days, respectively, similar to pre-COVID-19 results. In conclusion, for colorectal and lung cancer, the time to primary care referral was substantially prolonged during the first COVID-19 wave. In such crises, targeted primary care support is needed to maintain effective cancer diagnosis.

## 1. Introduction

The Coronavirus disease 2019 (COVID-19) pandemic has had a substantial impact on health care worldwide, altering the manner in which patients accessed care, health professionals provided care, and health care systems functioned [1,2,3]. As the first wave of COVID-19 struck Europe in March 2020, health care provision and government policy focused on care for patients with COVID-19 and the prevention of its transmission. Lockdowns were implemented, and usual care processes were interrupted or stopped, producing major impacts on care for patients with non-COVID-19 conditions, such as cancer [3,4,5]. This is evident from cancer diagnosis data from March to June 2020, with numbers of cancer diagnoses reportedly decreasing between 25–61% internationally [6,7,8].

Delay in recognition, referral, and diagnosis of cancer patients can have a substantial impact on the disease burden and prognosis [9,10]. Such delay is associated with later stages of cancer at diagnosis, more invasive treatments, greater impacts on patient lives and quality of life, and worse morbidity and mortality [9,10]. In countries with primary care-based healthcare systems, the diagnosis of cancer largely occurs through the general practitioner (GP), with previous studies demonstrating that over 80% of cancer patients in Western Europe are diagnosed after contact with their GP [11]. The first steps in the diagnostic process are integral for timely diagnosis and treatment; patients first recognize their symptoms, then present to their GP, who details these symptoms, investigates them if needed, and, if indicated, refers the patients to secondary care for diagnosis and treatment [12].

During the first COVID-19 wave, the opportunities to perform face to face consultations and physical examination were restricted and in GP consultations there was a dominant focus on COVID-19, both by patients and GPs [5,13,14,15,16]. Referral pathways were also affected, because of interruptions in routine investigations and overburdening of secondary care services [16]. Therefore, we hypothesize that the first COVID-19 wave had a substantial impact on the time to recognition, referral, and diagnosis of cancer patients presenting to primary care.

To enable the targeted prevention of delay in cancer diagnosis in future pandemics, detailed knowledge about the extent of such delay and its occurrence in specific populations is needed. To achieve this goal, we aim to map the impact of the first wave of the COVID-19 pandemic on the duration of the diagnostic pathway before and after primary care referral for symptomatic cancer patients in The Netherlands.

## 2. Materials and Methods

### 2.1. Study Design

We performed a retrospective cohort study. To build our cohort, we linked routine primary care data to cancer registry data collected in primary care practices and hospitals in The Netherlands. This study was reported to be in line with the Strengthening the Reporting of Observational Studies in Epidemiology (STROBE) Statement [17]. This research was reviewed by the institutional review board of the UMC Utrecht (21-144/C) and judged not to be subject to the Medical Research Involving Human Subjects Act of The Netherlands.

### 2.2. Data

Primary care data was obtained from The Intercity Data Network, which combines data from the dynamic primary care cohorts of five academic primary care networks: the Academic Network of General Practice at Amsterdam University Medical Centre; location VU Medical Center (ANH VUmc) and location Amsterdam Medical Center (AHA AMC), the Research Network Family Medicine (RNFM) Maastricht, the Academic General Practitioner Development Network Groningen (AHON), and the Julius General Practitioners’ Network Utrecht (JGPN). Together, they contain data from over 1.2 million adult primary care patients, which are representative of the Dutch population [18].

This data contains deidentified coded and free text data, including free text descriptions of patient symptoms, physical examinations, working diagnoses, and case management. The symptoms and diagnoses are coded according to the International Classification of Primary Care (ICPC-1) coding system.

Data from patients with diagnostic codes for cancer (colorectal, lung, and breast cancer, and melanoma) in the primary care dataset were linked to the Dutch National Cancer Registry (NCR) database by a third trusted party. The NCR is a national database which includes over 99% of Dutch cancer patients [19]. NCR data include patient demographics, tumour histology, stage and localisation of cancer, date of diagnosis, and subsequent treatment. The Intercity Data Network, the NCR, and the linkage process have been described previously [20,21,22].

### 2.3. Population

To select cancer patients for whom the primary care diagnostic interval (partially) overlapped with the first COVID-19 wave, we included patients from each database in the intercity network with a newly attributed ICPC-1 code for colorectal (code: D75), lung (R84), or breast (X76) cancer, or melanoma (S77.03) used after 1 March 2020. The diagnosis of cancer for each patient was confirmed by manual exploration of the free text data, thereby employing an established validation process described previously [20,21,22]. The first wave of the COVID-19 pandemic was defined as “from the first of March to the 30 June 2020,” informed by the National Institute for Public Health and the Environment (RIVM) [23]. We selected patients who presented to the GP with cancer related symptoms, who were then referred to secondary care, and whose primary care diagnostic pathway overlapped with all or part of the first COVID-19 wave. We ceased screening routine care data for such patients for each cancer type if there were no new inclusions for two consecutive months after the first COVID-19 wave. Patients diagnosed through screening programs, specialists, or emergency departments, or who were asymptomatic, were excluded. For melanoma, patients who had a melanoma resected by the GP (and thus, were not referred to secondary care) were not included.

Data for patients whose diagnostic interval served as the control period were collected in the previous Dickens project, as elaborated in the thesis of Nicole van Erp [22]. These data describe the duration of the diagnostic process of symptomatic patients diagnosed with colorectal, lung, and breast cancer, and melanoma, prior to the COVID-19 pandemic, from 1 January 2012 to 31 December 2015.

### 2.4. Data Collection

Information used to detail the time intervals of the diagnostic process in primary care was manually collected from the free text routine care data. All data collection was performed according to a data collection manual, which was developed in the previous Dickens project. We used identical methods for the pre-COVID-19 and COVID-19 cohorts [22]. Data collection was performed by the research team comprised of medical doctors and medical students. If there were uncertainties, these were discussed and resolved within the study group. All data required for analysis was entered into Castor, an electronic data capture system, using a standardized capture procedure.

The diagnostic time intervals employed in this study were developed from the Aarhus statement, described in Figure 1 [12]. The primary care interval (IPC) was defined as the period between the first contact with primary care for suspected cancer-related signs or symptoms (which could be in-person or through video/telephone consultation) and the date of referral to specialist care. The date of referral is generally explicitly mentioned in primary care data. If not, the moment when responsibility for patient care was transferred to specialist care was identified in the free text and selected. In the case of multiple referrals, the first referral relating to cancer-related symptoms was chosen.

The secondary care interval (ISC) was defined as the period between referral by the GP to secondary care and the histological cancer diagnosis, as retrieved from the NCR.

Patient demographics, clinical characteristics, signs and symptoms, and dates of first consultation and referral were collected from primary care databases. Comorbidities were manually collected for each patient in accordance with the methods of O’Halloran et al. [24]. Cancer alarm symptoms were assessed per cancer type according to predefined definitions, similar to those previously employed in the Dickens project to determine the pre-COVID-19 durations of IPC and ISC [22].

### 2.5. Analyses

Patient demographics and clinical characteristics were detailed using descriptive statistics. The durations of the diagnostic intervals, for each cancer population and its subgroups (patient and clinical characteristics) were calculated in days for each cohort. Durations were expressed in medians and interquartile ranges because of the expected non-parametric distribution. Consistent with previous studies, to define same day referrals as ‘one day,’ one day was added to all intervals. The number of GP consultations with cancer-related complaints (for the specific cancer type) in IPC was counted with a maximum of one consultation per day and included the initial and referral consultation. Mann–Whitney U tests were employed for comparisons of differences regarding duration and number of consultations (COVID-19 versus pre-COVID-19), and a *p*-value of less than 0.05 was considered statistically significant. IBM SPSS software version 27 (Microsoft, Chicago, IL, USA) was employed for data analysis.

## 3. Results

To assess the impact of the first wave of the COVID-19 pandemic on the duration of the diagnostic pathway before and after primary care referral for symptomatic cancer patients in The Netherlands, we performed a retrospective cohort study using routine primary care data linked to The Netherlands Cancer Registry.

We screened 3182 cancer patients for eligibility (IPC overlap the first COVID-19 wave). Overlap of the first COVID-19 wave with IPC was found for 415 cancer patients, and with ISC for 273 patients. The inclusion process is described in a flow diagram in the Appendix A. In the prior Dickens project, which describes the pre-COVID-19 period, IPC and ISC durations were determined for 979 cancer patients [22].

### 3.1. Population

Table 1 describes the patient demographics and clinical characteristics of the patients whose IPC or ISC overlapped with the first COVID-19 wave, and of patients included in the pre-COVID-19 study. The mean age at the first GP consultation was between 55 and 70 years, with the majority of patients (64–92%) having registered comorbid conditions.

### 3.2. Primary Care Interval (IPC)

As shown in Figure 2, the primary care interval was prolonged during the first COVID-19 wave for patients with colorectal and lung cancer, but not for patients with breast cancer and melanoma. For colorectal cancer patients, the median IPC duration increased by 39 days, from 5 days (IQR 1–29) pre-COVID-19, to 44 days (IQR 6–230, *p* < 0.01) during the first wave. For lung cancer patients, the median duration from first consultation to referral increased by 26 days, from 15 days (IQR 3–47) to 41 days (IQR 7–101.75, *p* < 0.01). For breast cancer and melanoma patients the median IPC duration remained 1 day, and the P75 value, indicating the value above which the 25% longest durations occur, shifted from 1 to 2 days (not statistically significant). This lack of relevant change for breast cancer and melanoma was observed in all subgroup analyses, with median durations remaining at 1 day in all subgroups (not shown in Table).

Subgroup analyses suggest differences in impact of the first COVID-19 wave on IPC duration for colorectal and lung cancer, as detailed in Table 2. For colorectal cancer, the observed increase in duration was the largest for females, young patients (<65), those with less than two comorbidities, and those with psychiatric comorbidity. For those with colorectal cancer and alarm symptoms (e.g., rectal bleeding), the median IPC duration increased from one day (IQR 1–18) pre-COVID-19 to one month during COVID-19 (IQR 13–169, *p* < 0.01). In contrast, for lung cancer, the largest absolute increase was observed for males, elderly patients (≥65), and those with more than two comorbidities or psychiatric comorbidity.

The number of cancer symptom related GP consultations in IPC—for all cancer types—were significantly increased during the first COVID-19 wave. For colorectal and lung cancer, the median number of consultations in IPC was 3 (IQR; 2–6) and 4 (IQR; 3–6), as compared to 2 (IQR; 1–4) (*p* value for change: <0.01) and 3 (IQR; 2–5) (*p* value for change: <0.01) pre-COVID-19. For breast cancer and melanoma, the median number of consultations remained at 1, but a larger proportion of patients required more than one consultation (*p* value for change: both <0.05).

### 3.3. Secondary Care Interval (ISC)

The ISC duration was only prolonged for breast cancer, increasing from a median of 3 days (IQR 2–7) before COVID-19, to 6 days (IQR 3–9, *p* < 0.01) during the first COVID-19 wave. ISC durations during the first COVID-19 wave for colorectal and lung cancer and melanoma were 17.5 (IQR 9–52), 18 (IQR 7–40), and 9 (IQR 3–44) days, respectively, which is similar to the pre-COVID-19 period. Only for lung cancer, the subgroup analyses showed differences; a significant increase in ISC duration for those with psychiatric co-morbidity, from a median of 16 days (IQR 8–30.5) pre-COVID-19 to 40 days during COVID-19 (IQR 14–49). Details on ISC duration before and during the first COVID-19 wave are available in the Appendix A.

## 4. Discussion

### 4.1. Main Findings

These results describe substantial delays in primary care for colorectal and lung cancer patients during the first COVID-19 wave, particularly among the subgroups. For colorectal and lung cancer, the median primary care diagnostic interval during the first COVID-19 wave was 39 and 26 days longer than the pre-COVID-19 interval, respectively. Subgroup differences were suggested, affecting different groups for colorectal and lung cancer. The impact of COVID-19 on the secondary care interval was minimal, with only a small increase of three days observed for breast cancer.

The delays observed for colorectal and lung cancer in primary care are worrying, given the extent of these delays and the tendency of these tumor types for rapid progression. Sud et al. studied the effect of a 2-month delay for major cancer types in a modeling study that showed an 11% and 7% worsening in 10-year survival rates for lung and colorectal cancer (estimated for a 65-year old patient) [9]. Especially when considering the observed increase in duration of two to six months in P75 values (longest: 25%) for the diagnostic interval time in our study, the potential impact of the COVID-19-related delay on cancer burden and prognosis is worrisome.

There were concerns that the impact of the COVID-19 pandemic would disproportionally affect vulnerable patients [13]. Our results suggest that this may be true for colorectal and lung cancer patients with psychiatric comorbidities, for whom the impact of COVID-19 on the duration of the primary care pathway was observed to be twice as severe. Another worrying finding is that patients with colorectal and lung cancer presenting with alarm symptoms were also at increased risk of delay during the first COVID-19 wave. For both these patient populations, targeted delay prevention appears warranted in the case of future pandemics or similar disruptions to health systems.

### 4.2. Implications for Practice

The finding that delays during the first COVID-19 wave occurred in colorectal and lung cancer patients, and hardly at all in melanoma and breast cancer, reflects the known challenges in detecting ‘hard-to-recognise’ cancers. While the symptoms of melanoma and breast cancer are generally clear, those of lung and colorectal cancer tend to be less specific. During the first COVID-19 wave, recent studies suggest that recognizing and acting upon less specific symptoms was increasingly challenging [13,16,25], particularly for lung cancer, since its symptoms (e.g., coughing) were considered a reason to shift to the use of video or telephone consultation instead of physical consultation during the first COVID-19 wave. This additional challenge in recognizing and referring cancer is reflected not only in a longer duration of the primary care pathway, but also in the additional consultations needed prior to referral. This is likely to be partially related to the increased use of telehealth during this period, resulting in many initial consultations through this medium, requiring a secondary consultation to facilitate a physical examination and a more detailed assessment [25]. However, this alone is unlikely to be responsible for the observed delay. On a patient level, recent studies described patient-related barriers to being referred to specialist services during this period, including a reluctance to ‘make a fuss,’ and concerns of overburdening the health system or being infected with COVID-19 [16,25]. On a system level, health professionals described challenges in accessing investigations and communicating with and referring to secondary care [25]. Such barriers for referral are likely to be accentuated for cancer types that generally present with alarm symptoms that are less distinct and for which referral pathways are more complex.

The secondary care diagnostic pathway did not appear to have experienced the same extensive delays during the first COVID-19 wave as those seen in the primary care phase. This is likely to be the effect of the well-developed and tailored nature of secondary care cancer pathways and their prioritization of cancer care in times of increased pressure on health care.

Future directions: our findings show that the impact of the first wave of the COVID-19 pandemic on the diagnostic pathway of symptomatic cancer was the largest in primary care. The mechanisms underlying primary care delay seem multifactorial and inconsistent between cancer types and subgroups. To reduce the impact of future outbreaks on effective cancer diagnosis, improving understanding of the mechanisms leading to delay would facilitate the development of targeted support for the detection and referral of cancer patients by GPs.

### 4.3. Strength and Limitations

This study employed routine care data from patients throughout The Netherlands for both the prior Dickens project and the current study. The main strength of such data is the large and diverse population from which the data is obtained, as well as the detailed clinical nature of the data. The availability of free text data, which can be used as a prospectively collected transcribed verbatim summary of daily practice consultations, allows very rich and detailed information to be used to determine milestones which mark the beginning and end of the primary care phase of the diagnostic pathway. It also enables the detailed registration of a broad range of symptoms and characteristics, which are not always registered in coded data. As symptoms are registered at the time of their occurrence, recall bias is eliminated. This rich primary care dataset includes over 1.2 million adult primary care patients, from different geographic and socioeconomic regions, which was then linked with the Netherlands Cancer Registry (NCR) database. Linkage to the NCR allows for the use of detailed and reliable information, such as the validation of cancer diagnosis and its date. Data from the pre-COVID-19 period (2012–2015) were used as a comparator. They originated from the same datasets, and we employed the same methods and measures for data collection, providing methodological consistency in order to compare results.

Several limitations of this study should be taken into account. The numbers of cancer cases occurring in this study are substantially fewer than those in the prior Dickens study, which was used as the comparator. This is largely due to the inclusion criteria, which demand that presentation to the GP occurs during the first COVID-19 wave, which is a much shorter timeframe than that used in the previous Dickens study. Despite the dataset including over 1.2 million patients, only a small proportion of patients were diagnosed with these cancer types and had exhibited their diagnostic periods within this timeframe. We expect to have included all eligible patients within this inclusion period because of continued screening for patients up to one year after the end of the first COVID-19 wave. However, it is possible that, in the event that cancer was diagnosed more than one year after the first COVID-19 wave, we missed some patients with extensive delays.

Routine care data contains complexities, errors, and intricacies. This data is entered by many different health practitioners while—and for the purpose of—providing clinical care for patients. Thus, data collection was very time consuming and can be subject to interpretation. For optimal yield and the prevention of error and subjectivity, data collection was done according to a set framework, and was performed by different researchers with clinical experience, such as medical students or doctors. Similar to the prior Dickens project, if there were uncertainties or missing information in the data that made the diagnostic pathway of the patient unclear, these patients were excluded. Despite this extensive effort, our data may be incomplete and subject to misinterpretation. Routine care data can be incomplete or sub-optimally coded for research purposes. Consequently, missing diagnostic codes could have occurred; i.e., for some patients, the right ICPC code for the cancer diagnosis might not (yet) have been attributed by the GP [20]. This would lead to missing cancer patients in our dataset. Additionally, if there were uncertainties about the cancer diagnostic processes (i.e., the presence of cancer related symptoms, or when a referral occurred), these patients were excluded. A total of 555 patients were excluded because of an ‘unclear diagnostic pathway’. Fortunately, the number of patients which could be included were consistent with our original estimates, which were based on the previous Dickens project. We believe the total numbers included are representative and sufficient to robustly demonstrate clinically relevant differences in the diagnostic periods in comparison with the pre-COVID-19 data. Moreover, by adding NCR data to the primary care data required linkage using pre-approved third trusted-party employment and privacy procedures, ensuring the non-reducibility of the data. Such procedures, similar to those in preceding projects, lead to loss of patients, which is generally thought to be a non-selective loss [22].

It should be noted that our analyses of the impact of the COVID-19 outbreak do not include the delay in the diagnostic process which may have occurred before presentation to primary care, in other words, the patient interval from the time of first symptom recognition to consulting primary care. Recent findings suggest that there was widespread avoidance of primary care during the first COVID-19 wave, with reductions of up to 34% in patient presentations with cancer-related symptoms during this period [14,26,27,28]. This delay due to care avoidance is likely to add to the delay observed in our study.

Moreover, our findings only describe the impact of the first COVID-19 wave and not the impact during the full COVID-19 pandemic. The impact of the first wave on both health care performance and patient behavior is likely to be larger during the first wave than during consecutive waves [3,27,28,29].

Finally, it should be taken into account that—for colorectal and breast cancer—a national screening program exists in The Netherlands, which was halted because of COVID-19 halfway through March 2020. Additionally, we did not include patients visiting the emergency room. Therefore, our findings are not representative of all new cancer patients, and changes in these other pathways may have influenced the selection of patients visiting primary care with symptoms [30]. Given our focus only on symptomatic patients presenting to primary care during the relatively short first wave of COVID-19, we do not expect this to have impacted our findings.

## 5. Conclusions

This study demonstrates that time to referral in primary care was substantially prolonged during the first COVID-19 wave for patients with colorectal or lung cancer. This impact seems to vary for subgroups, and it is inconsistent between cancer types. For these ‘hard-to-diagnose-cancer-types’ that exhibit less distinct referral criteria and pathways, delay also occurred in patients presenting with alarm symptoms. For patient populations at risk of pandemic-related delays in primary care, targeted support for detection and referral by GPs seems warranted to ensure effective cancer diagnosis in future pandemics or similar crises.

## Figures and Tables

**Figure 1 cancers-15-01462-f001:**
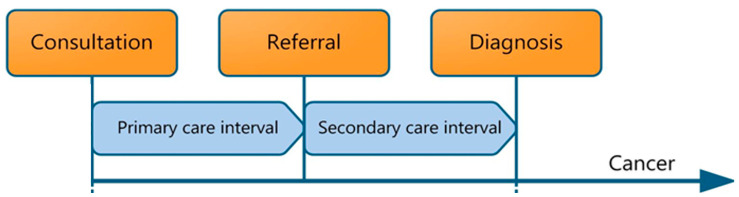
Primary care cancer diagnostic intervals included in the pathway from time of first symptom presentation to diagnosis, including the primary care interval (IPC) and the secondary care interval (ISC), as detailed in the Aarhus statement [12].

**Figure 2 cancers-15-01462-f002:**
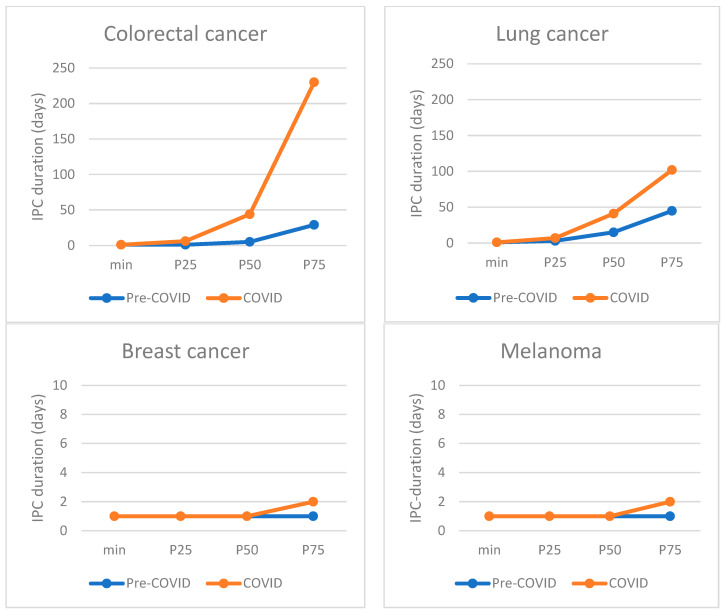
Distribution of primary care interval (IPC) durations for each cancer type, comparing IPC durations pre-COVID-19 with those during COVID-19. P50 marks the median, P25 the 25th percentile, and P75 the 75th percentile.

**Table 1 cancers-15-01462-t001:** Patient demographic and clinical characteristics per cancer type for pre-COVID-19 (2012–2015) and the first COVID-19 wave (March–June 2020). Registered comorbidity was defined in accordance with O’Halloran et al. [24].

Cancer Type	Colorectal		Lung		Breast		Melanoma	
Time Period	Pre-COVID-19	COVID-19	Pre-COVID-19	COVID-19	Pre-COVID-19	COVID-19	Pre-COVID-19	COVID-19
**Primary Care Interval**								
**N**	313	110	236	118	306	140	124	47
Female—*n* (%)	154 (49)	66 (60)	104 (44)	62 (53)	306 (100)	140 (100)	61 (49)	28 (60)
Age at first GP consultation—mean (SD)	69.5 (12.5)	68.4 (14.4)	68.6 (11.1)	66.4 (10.9)	57.5 (18.2)	58.9 (17.4)	55.0 (17.0)	60.7 (18.1)
Registered comorbidity—*n* (%)	287 (92)	90 (82)	218 (92)	96 (81)	241 (79)	90 (64)	85 (69)	31 (66)
Psychiatric comorbidity—*n* (%)	79 (25)	20 (18)	48 (20)	27 (23)	52 (17)	42 (30)	28 (20)	8 (17)
**Secondary Care Interval**								
**N**	259	62	197	75	256	110	106	26
Female—*n* (%)	130 (50)	33 (53)	87 (44)	42 (56)	256 (100)	110 (100)	52 (49)	14 (54)
Age—mean (SD)	69.1 (12.4)	67.8 (14.7)	68.6 (10.6)	67.0 (9.6)	57.8 (17.4)	57.3 (17.3)	58.3 (15.2)	65.0 (16.8)
Registered comorbidity—*n* (%)	236 (91)	52 (84)	182 (92)	65 (87)	202 (79)	74 (67)	77 (73)	23 (89)
Psychiatric comorbidity—*n* (%)	66 (26)	13 (21)	40 (20)	14 (19)	44 (17)	32 (29)	19 (18)	2 (8)

**Table 2 cancers-15-01462-t002:** Impact of first COVID-19 wave on duration of the primary care interval—total and subgroups.

Cancer Type and COVID-19 Impact	Colorectal Cancer				Lung Cancer			
	Pre COVID-19	1st Wave COVID-19	*p* for Diff.	Absolute Increase Median Duration (Days)	Pre COVID-19	1st Wave COVID-19	*p* for Diff.	Absolute Increase Median Duration (Days)
	Median IPC Duration in Days *(IQR)*				Median IPC Duration Days *(IQR)*			
**Total** ** * Number of patients (N) * ** ** N = CPC: 313 CC: 110 LPC: 166 LC: 118 **	**5** ** * (1–29) * **	**44** ** * (6–230) * **	** <0.01 **	**39**	**15** ** * (3–47) * **	**41** ** * (7–102) * **	**<0.01**	**26**
**Gender** **Male** —N: CPC: 159 CC: 44 LPC: 132 LC: 56	**3** * (1–25) *	**30** * (2–177) *	<0.01	27	**13** * (2–40) *	**46** * (7–98) *	<0.01	33
**Female**—N: CPC: 154 CC:66 LPC: 104 LC: 62	**7** * (1–30) *	**66** * (8–252) *	<0.01	59	**18** * (5–50) *	**40** * (6–107) *	0.04	22
**Age** **<65** —N: CPC: 100 CC: 37 LPC: 83 LC: 50	**4** * (1–24) *	**67** * (12–240) *	<0.01	63	**15** * (5–51) *	**31** * (7–92) *	0.05	16
**≥65**—N: CPC: 213 CC: 73 LPC: 153 LC: 68	**6** * (1–32) *	**37** * (4–231) *	<0.01	31	**15** * (3–44) *	**47** * (7–105) *	<0.01	32
**Comorbidities present** **<2**—N: CPC: 60 CC: 44 LPC: 53 LC: 43	**2** * (1–15) *	**57** * (2–231) *	<0.01	55	**10** * (1–24) *	**29** * (8–99) *	<0.01	19
**≥2**—N: CPC: 247 CC: 66 LPC: 177 LC: 75	**6** * (1–31) *	**42** * (8–231) *	<0.01	36	**20** * (5–51) *	**46** * (6–108) *	<0.01	27
**Psychiatric comorbidity** **Yes** —N: CPC: 79 CC: 20 LPC: 48 LC: 27	**5** * (1–19) *	**70** * (22–259) *	<0.01	65	**19** * (5–50) *	**55** * (11–123) *	<0.01	36
**No**—N: CPC: 230 CC: 90 LPC: 187 LC: 91	**5** * (1–33) *	**27.5** * (4–216) *	<0.01	22.5	**15** * (2–44) *	**32** * (5–92) *	<0.01	17
**Alarm symptom at first consult** **Yes** —N: CPC: 169 CC: 45 LPC: 188 LC: 45	**1** * (1–18) *	**37** * (13–169) *	<0.01	36	**18** * (1–18) *	**37** * (2–169) *	0.6	19
**No**—N: CPC: 144 CC: 65 LPC: 188 LC: 101	**11.5** * (1–35) *	**52** * (13–252) *	<0.01	40	**18** * (6–50) *	**48** * (11–108) *	<0.01	30
**GP consult frequency**—prior year **<5**—N: CPC: 130 CC: 56 LPC: 120 LC: 71	**4** * (1–26) *	**25** * (4–189) *	<0.01	21	**11** * (2–37) *	**39** * (9–88) *	<0.01	28
**≥5** —N: CPC: 103 CC: 54 LPC: 66 LC: 47	**6** * (1–29) *	**50** * (10–232) *	<0.01	44	**18.5** * (6–50) *	**45** * (5–108) *	0.04	26.5

(N): number of patients; CPC: colorectal cancer pre-COVID-19; CC: colorectal cancer 1st wave COVID-19; LPC: lung cancer pre-COVID-19; LC: lung cancer 1st wave COVID-19. If numbers do not add up to expected total, this is due to missing (left out) information. *p*. for diff.: *p*-value for difference in IPC durations pre-COVID-19 and during first COVID-19 wave. No significance testing for differences in COVID-19 impact between subgroups within cancer populations was performed because of low N. IQR: interquartile range. Periods: pre-COVID-19: 2012–2015; first COVID-19 wave: March–June 2020. Comorbidity was defined in accordance with O’Halloran et al. [24].

## Data Availability

The de-identified participant data used for the analysis can be disclosed by the corresponding author upon reasonable request.

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
