# Peer review of "Impact of the COVID-19 Outbreak—Delayed Referral of Colorectal and Lung Cancer in Primary Care: A National Retrospective Cohort Study"

_cancers, 2023, doi:10.3390/cancers15051462_

Round 1

Reviewer 1 Report

1 -There are several English mistakes. Avoid ambiguous sentences. The manuscript should be edited.

2-Authors needs to mention the limitations and future direction of this study.

3- M&M sections require more precise descriptions. All your methods should have appropriate citation of references. Check and revise them properly.

4-Add clearly the hypothesis, aims and goals of this work to the last paragraph to your introduction.

5-Please add a starting paragraph to the results section to briefly introduce the topic, your goals and hypothesis and a short summary of what you did in this work. Many readers start reading the manus from the results and it should be understandable.  

6- All the abbreviations should be explained when used the first time in the manuscript. In addition, if you can avoid any of the abbreviations, it is preferred to write only full text

Reviewer 2 Report

In this study, the authors investigated the impact of the first wave of the COVID-19 pandemic on the duration of the diagnostic pathway before and after primary care referral for symptomatic cancer patients in the Netherlands. The results show that time to referral in primary care was substantially prolonged during the first COVID-19 wave for patients with colorectal- or lung cancer. The authors suggest that, for these patient populations at risk of pandemic-related delays in primary care, targeted support for detection and referral by GPs is needed to ensure effective cancer diagnosis in future pandemics or similar crises. The manuscript is very well prepared. The results and suggestions of the study are very important to future cancer care and treatment in pandemics or similar sociomedical crises, and should be published.

Author Response

We would like to thank the reviewer for the supportive comments.

Reviewer 3 Report

This article is interesting and clinically relevant.

Authors discussed several methodological limitations.

However, I think, there is one more limitation. COVID-19 pandemic began in Europe in February of 2020 (maybe even March) and last till today. The first wave they investigated was very short. In this time, people had a lot of anxitey caused by the governement, partly TV or newspapers- with a consequence that they did not want to leave their homes and avoided each visit by physicians. Some practices and ambulances closed or strongly reduced the number of patients which they treated. But some months later, already starting with July 2020 and especially in 2021 and 2022 the situation was less dramatical. People started to meet again, they gone to physicians and so on.  Let me say, this is not the impact of COVID-19 pandemic, what authors presented in their work, but rather the effect observed in the first COVID-19 wave. Maybe this is also the efefct of anxiety or effect of anto-pandemic measures. This should be discussed, at least in limitations, and maybe even reflected in the study title.

Author Response

Dear reviewer, thank you for this valuable reflection and suggestion.

We fully agree that the initial outbreak is likely to have had a far greater impact than the sequential waves.

We have added this reflection to our discussion section and we altered the title of the manuscript to; “Impact of the COVID-19 Outbreak – Delayed Referral of Colorectal and Lung Cancer in Primary Care; a National Retrospective Cohort Study “.

The discussion section now states:

“ ….  our findings only describe the impact of the first COVID-19 wave and not the impact during the full COVID-19 pandemic. The impact of the first wave on both health care performance and patient behaviour is likely to be larger during the first wave than during consecutive waves”.